# DNA-binding site II is required for RAD51 recombinogenic activity in *Arabidopsis thaliana*

Valentine Petiot, Charles I White, Olivier Da Ines

**Homologous recombination is a major pathway for the repair of DNA double strand breaks, essential both to maintain genomic integrity and to generate genetic diversity. Mechanistically, homologous recombination involves the use of a homologous DNA molecule as a template to repair the break. In eukaryotes, the search for and invasion of the homologous DNA molecule is carried out by two recombinases, RAD51 in somatic cells and RAD51 and DMC1 in meiotic cells. During recombination, the recombinases bind overhanging single-stranded DNA ends to form a nucleoprotein filament, which is the active species in promoting DNA invasion and strand exchange. RAD51 and DMC1 carry two major DNA-binding sites—essential for nucleofilament formation and DNA strand exchange, respectively. Here, we show that the function of RAD51 DNA-binding site II is conserved in the plant, Arabidopsis. Mutation of three key amino acids in site II does not affect RAD51 nucleofilament formation but inhibits its recombinogenic activity, analogous to results from studies of the yeast and human proteins. We further confirm that recombinogenic function of RAD51 DNA-binding site II is not required for meiotic double-strand break repair when DMC1 is present. The Arabidopsis *AtRAD51-II3A* separation of function mutant shows a dominant negative phenotype, pointing to distinct biochemical properties of eukaryotic RAD51 proteins.**

## Introduction

Homologous recombination (HR) is a DNA repair process found in all forms of life that is pivotal for maintaining genomic integrity and ensuring genetic diversity (Ranjha et al, 2018; Wright et al, 2018). In somatic cells, HR is used to repair DNA lesions, such as DNA double-strand breaks (DSBs), caused by environmental or endogenous factors and to restore stalled or collapsed DNA replication forks (Ranjha et al, 2018; Wright et al, 2018). HR is also essential for sexual reproduction in most studied eukaryotes, where it promotes accurate chromosome segregation and enables the exchange of genetic information between parental DNA molecules, thereby generating genetic diversity among meiotic products (Hunter, 2015; Wang & Copenhaver, 2018).

HR uses an intact homologous DNA molecule as a template for copying and restoring the information at the break. The central step of HR is thus search for a homologous DNA template and its invasion by the ends of the broken DNA molecule. In eukaryotes, these key steps are catalyzed by the RecA recombinase homolog RAD51 in somatic cells, and, in general, RAD51 and DMC1 in meiotic cells (Brown & Bishop, 2014; Crickard & Greene, 2018; Emmenecker et al, 2023).

Once a DSB is formed, DSB ends are recognized and processed to generate long 3′-OH single-stranded DNA overhangs (ssDNA) (Cejka & Symington, 2021). The ssDNA overhangs are then coated by the ssDNA-binding protein RPA (Replication Protein A), stabilizing and protecting them from nucleases and the formation of secondary structures (Chen et al, 2013; Chen & Wold, 2014). RPA is subsequently displaced by RAD51 in somatic cells, or by RAD51 and DMC1 in meiotic cells, forming a right-handed, helical nucleofilament on the ssDNA flanking the DSB (Brown & Bishop, 2014; Crickard & Greene, 2018; Emmenecker et al, 2023). This helical nucleoprotein filament is the active molecular machinery that performs the homology search and catalyzes the invasion of a homologous double-stranded DNA (dsDNA) donor sequence. DNA strand invasion of the dsDNA template by the recombinase-ssDNA nucleofilament generates a displacement loop (D-loop), which can then be extended through DNA synthesis. The resulting recombination intermediate will eventually be resolved through one of several different enzymatic pathways, leading to separation of the recombining DNA molecules and restoration of DNA integrity.

Hence in vivo, RAD51 or DMC1 recombinases assemble to form an active nucleoprotein filament on RPA-coated ssDNA that will catalyze strand exchange. This two-step process is tightly regulated by a number of positive and negative cofactors but also strongly relies on the biochemical properties of the recombinases (Zelensky et al, 2014; Kowalczykowski, 2015; Emmenecker et al, 2023; Ito et al, 2024). RecA-family recombinases carry two separate DNA-binding sites (site I and II): site I is a high affinity DNA-binding site essential for polymerization on ssDNA and site II is considered a low-affinity binding site promoting interaction between the ssDNA-nucleoprotein

---

Institut Génétique, Reproduction et Développement (iGReD), CNRS UMR 6293, INSERM U1103, Université Clermont Auvergne, Clermont-Ferrand, France

Correspondence: Olivier.da_ines@uca.fr

filament and a second dsDNA molecule (Müller et al, 1990; Mazin & Kowalczykowski, 1996; Cloud et al, 2012; Prentiss et al, 2015; Xu et al, 2017; Ito et al, 2020). Site I is oriented towards the inside of the nucleoprotein filament and contains two conserved loops, Loop 1 and Loop 2. Both loops play distinct catalytic roles in the DNA strand exchange reaction (Xu et al, 2017; Ito et al, 2020) and recent evidence suggests that they have a critical role in the differing tolerance to DNA mismatches of the RAD51 and DMC1 recombinases during the invasion step (Steinfeld et al, 2019; Luo et al, 2021; Xu et al, 2021). Although studied in less detail, site II appears critical for the catalytic activity of recombinases assembled on ssDNA during homology search and strand exchange (Mazin & Kowalczykowski, 1996; Kurumizaka et al, 1999; Cloud et al, 2012; Prentiss et al, 2015; Ito et al, 2020). Initially characterized in *Escherichia coli* RecA, site II in RecA comprises the two positively charged residues: Arg243 (R243) and Lys245 (K245) (Kurumizaka et al, 1999). This is completed by a third positively charged residue, Arg227 (R227). These three residues form a basic patch on the groove of the helical filament (Cloud et al, 2012). In *Saccharomyces cerevisiae* Rad51, these three positively charged residues, essential for the function of site II correspond to Lys361 (K361) and Lys371 (K371), completed by Arg188 (R188) located in Walker A domain (Cloud et al, 2012). In human RAD51 they correspond to R303 and K313 (or Q313) completed with Walker A R130 (Mason et al, 2019). In both *S. cerevisiae* and human, mutating these three residues to alanine resulted in the separation-of-function mutant Rad51-II3A, which retains the ability to bind ssDNA and form the nucleoprotein filament, but is defective in strand invasion and D-loop formation (Cloud et al, 2012; Mason et al, 2019). Importantly, analysis of this mutant demonstrated that it is the DMC1 protein that catalyzes homology search and strand exchange during meiotic recombination, with RAD51 relegated to a supporting role (Cloud et al, 2012).

Relatively little is known concerning the structural features of the RAD51 recombinase in plants and notably, whether the function of DNA-binding sites I and II is conserved has not been demonstrated. We have previously shown that fusing a GFP to the C-terminal of *Arabidopsis thaliana* RAD51 impairs its recombinogenic activity without impairing nucleoprotein filament formation, a phenotype equivalent to that of the *S. cerevisiae* and human Rad51-II3A mutants (Da Ines et al, 2013; Singh et al, 2017). In vitro work has confirmed that the GFP fusion impacts the second-DNA–binding capacity (Kobayashi et al, 2014); however, it remains uncertain whether this is specifically through inactivation of the site II domain.

To clearly establish the nature and function of RAD51 DNA-binding site II in *A. thaliana*, we have generated an Arabidopsis *AtRAD51-II3A* mutant (mutation of the three conserved site II residues) and characterized its activity in vivo. We show that AtRAD51-II3A assembles at DNA break sites in both mitotic and meiotic cells, but has lost its recombinogenic activity. This severely impacts somatic DSB repair and recombination but does not impact meiotic recombination when DMC1 is present. Our data thus demonstrate that the function of RAD51 site II is conserved in Arabidopsis. This phenotype is equivalent to that of the Arabidopsis RAD51-GFP proteins and confirms that the recombinogenic activity carried by DNA-binding site II is not essential for meiotic DSB repair when DMC1 is present. Importantly, our data also show a dominant

negative effect of the AtRAD51-II3A protein, again equivalent to the Arabidopsis RAD51-GFP (Da Ines et al, 2013). Similar observations have been inferred in human (Saayman et al, 2023) but this contrasts clearly with results from *S. cerevisiae* (both Rad51-II3A and Rad51-GFP) (Cloud et al, 2012; Da Ines et al, 2013; Mason et al, 2019; Waterman et al, 2019). Our results thus clearly confirm the conservation of the structure and function of *A. thaliana* RAD51 DNA-binding site II and point to differing biochemical properties of eukaryotic RAD51 proteins.

# Results

### Construction of RAD51 site II mutant

To characterize the molecular functions of Arabidopsis RAD51 DNA-binding site II, we generated a mutant allele of Arabidopsis RAD51 based on the *S. cerevisiae* and human Rad51-II3A mutants (Cloud et al, 2012; Mason et al, 2019). We first compared Arabidopsis RAD51 protein sequence with that of *S. cerevisiae* and human RAD51 (Fig 1). Arabidopsis RAD51 (AtRAD51) displays high identity with budding *S. cerevisiae* and human RAD51, showing respectively 54% (68% similarity) and 68% (84% similarity) identity to the *S. cerevisiae* and human proteins. The structure and domains of all three RAD51 are strongly conserved with high identity in the Walker A and B motifs and DNA-binding sites I and II (Fig 1A and B). *S. cerevisiae* Rad51 carries an N-terminal tail that is absent in both Arabidopsis and human RAD51.

In *S. cerevisiae*, site II includes the three essential positively charged residues Arg188 (R188), Lys361 (K361) and Lys371 (K371). These correspond to R130, R303, and K313 (or Q313) in human (Fig 1). The three aminoacids are conserved in Arabidopsis (Fig 1B; R133, R306, and K316). We used the AlphaFold structure prediction of Arabidopsis RAD51 (AF-P94102-F1 model; Fig 2A) to compare its structure and arrangement with that of *S. cerevisiae* and *Homo sapiens* RAD51 (crystal structure PDB ID:1szp and PDB ID:5h1c, respectively) (Conway et al, 2004; Xu et al, 2017) (Fig 2B and C). The structure model predicted by AlphaFold gives a very confident model with very high per-residue confidence score (pLDDT > 90; blue color) for most residues (average pLLDT 91.38, Fig 2A top). The predicted aligned plot also shows very accurate prediction of the relative position of most residues within the structure (Fig 2A bottom). Only the first 24 residues (N-terminal tail) show a poorly predicted structure (pLDDT < 50, or 70 > pLDDT > 50), as well as aminoacids 276–286 (70 > pLDDT > 50). Interestingly, the latter are embedded within the DNA-binding loop 2 of RAD51 site I, which is suggested to play an important role in ssDNA binding and RAD51 mismatch sensitivity (Xu et al, 2017, 2021; Steinfeld et al, 2019; Ito et al, 2020; Luo et al, 2021). The predicted model strongly overlaps with the crystal structures of *S. cerevisiae* and Human RAD51 (Fig 2B and C; PDB ID:1szp and PDB ID:5h1c, respectively) (Conway et al, 2004; Xu et al, 2017). In particular, the three positively charged residues of AtRAD51 site II very robustly overlap with those in *S. cerevisiae* and human RAD51, all directed towards the outside. This prediction strongly supports the conservation of DNA-binding site II in Arabidopsis.

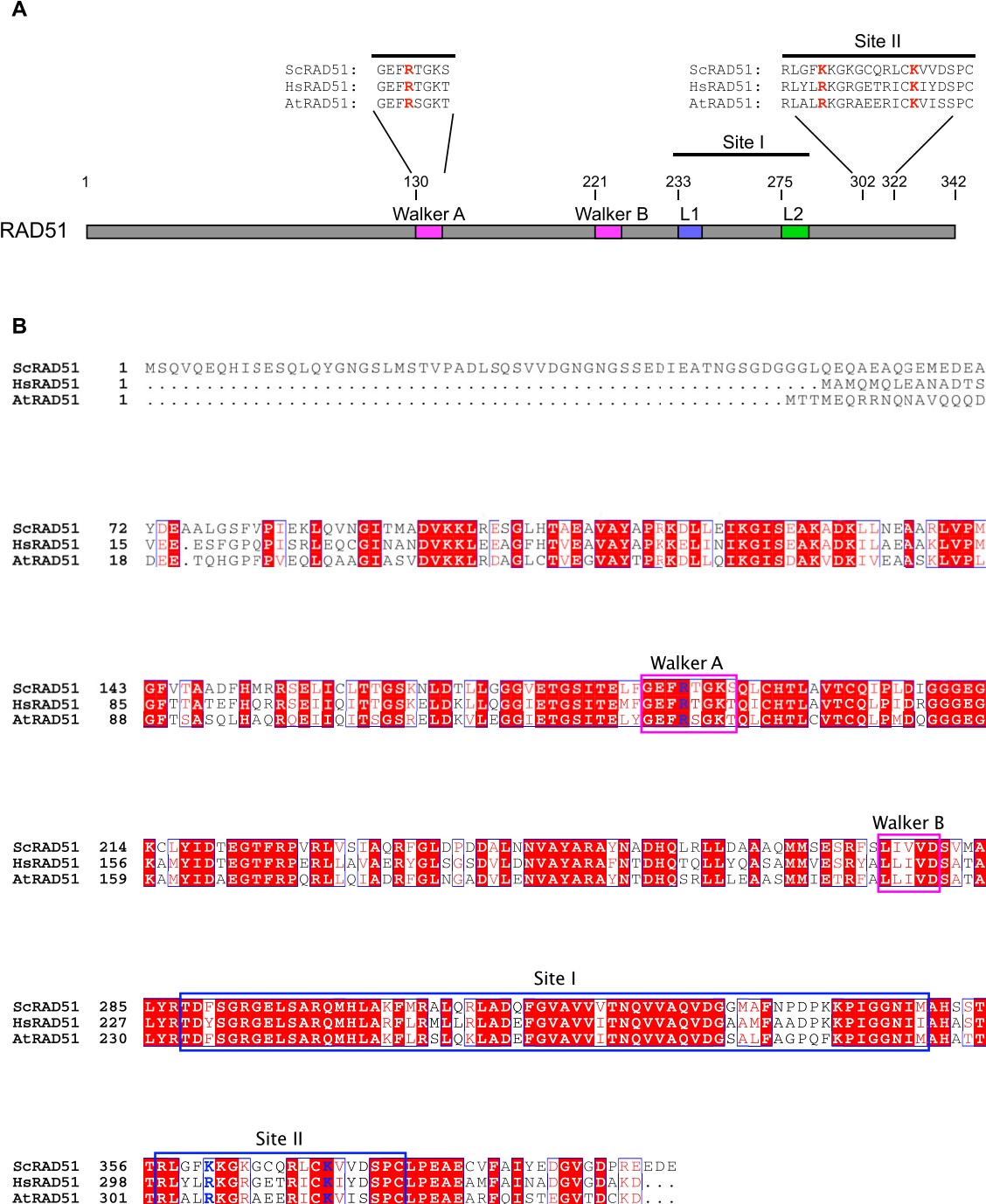

**Figure 1. AtRAD51 schematic structure and sequence.**
**(A)** Schematic representation of the domain structure of AtRAD51. The three essential amino acids mutated into alanine in the AtRAD51-II3A are displayed in red (R133, R306, and K316). R133 is located in the Walker A domain, whereas R306A and K316A are located in the DNA-binding site II. **(B)** Full alignment of *Saccharomyces cerevisiae*, *Homo sapiens*, and *Arabidopsis thaliana* RAD51 amino acid sequence. Mutated amino acids in the AtRAD51-II3A are written in blue. Walker (A and B) domains are outlined in magenta and DNA-binding sites (I and II) are outlined in blue. Conserved amino acids are highlighted in red, semi-conserved amino acids written in red, and non-conserved aa written in black.

We mutated these three residues to alanine to make *AtRAD51-II3A*. The *AtRAD51-II3A* sequence was placed under the control of the RAD51 promoter and this construct was introduced into Arabidopsis RAD51/*rad51-1* heterozygous plants. Among the primary transformants obtained, two (T1-1 and T1-2) were homozygous for the *rad51-1* allele and both were fertile, demonstrating that expression of the RAD51-II3A allele restores fertility of the *rad51* mutant.

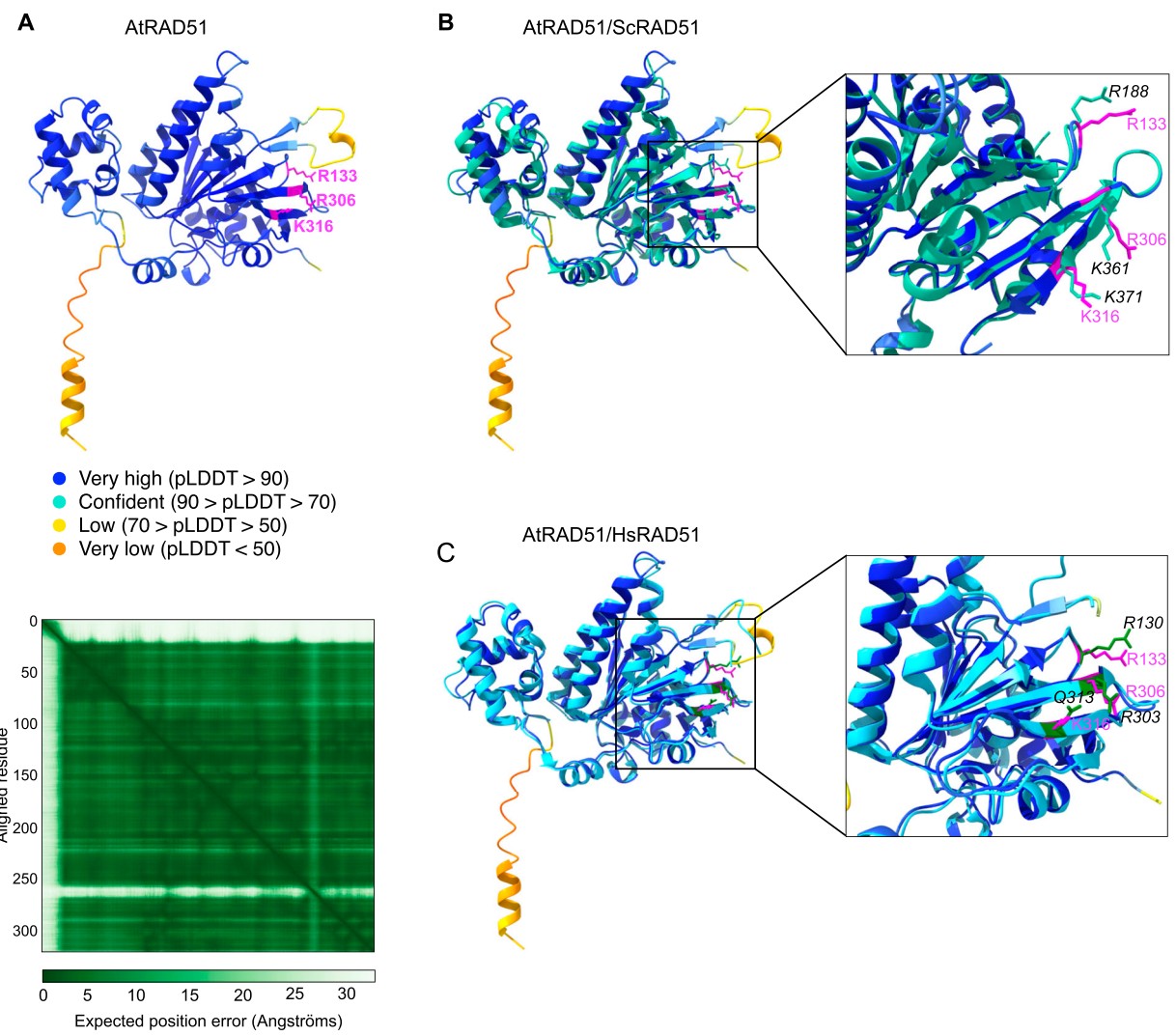

**Figure 2. 3D Structure model of AtRAD51.**
**(A)** Ribbon representation of the AlphaFold structure prediction of Arabidopsis RAD51 (AF-P94102-F1 model, top) and the corresponding Predicted Aligned Error output (bottom). The three essential amino acids in site II are labelled in magenta, and their position and name is written next to them (in magenta). In the Predicted Aligned Error plot, dark green tile corresponds to good prediction (low expected position error in Ångströms), whereas light green indicates low prediction (high error). **(B, C)** Superimposition of the AlphaFold proposed model structure of AtRAD51 and the crystalized structure of (B) *S. cerevisiae* (Turquoise; PDB ID:1szp) or (C) Human (Cyan; PDB ID:5h1c) RAD51. A close-up view of the region comprising the site II essential amino acids is displayed. **(B, C)** Amino acids are shown in magenta for AtRAD51, turquoise (black font) for ScRad51 (B), and cyan (black font) for human RAD51 (C). For better visualization of the site II essential amino acids, model structures in the close-up views have been slightly rotated and some residues hidden, compare to the views in the square.

## AtRAD51-II3A focus formation in response to DNA damage

Next, we sought to test whether AtRAD51-II3A retains ssDNA-binding activity in vivo. We thus analyzed AtRAD51-II3A focus formation upon DNA damage in somatic cells as a proxy for RAD51 nucleo-filament formation (hence ssDNA-binding at DNA break sites). We performed RAD51 immunofluorescence staining on root tip nuclei of 5-d-old seedlings treated or not with 30 $\mu$M mitomycin C (MMC; Fig 3). MMC induces DNA interstrand crosslink adducts and in turn, DNA strand breaks that are repaired by RAD51-dependent homologous recombination. Predictably, no or very few foci were detected in root tip nuclei of non-treated WT seedlings (Fig 3A),

whereas numerous RAD51 foci were detected 2 or 8 h after treatment with MMC (Fig 3B and C and dataset 1). The fraction of cells showing RAD51 labelling is relatively similar 2 or 8 h after treatment (around 25–30%; Fig 3D and E and dataset 1). In contrast, the number of RAD51 foci per nucleus significantly increases, with labelled nuclei showing 1–10 foci after 2 h treatments, and 1 to more than 20 foci after 8 h of treatment (Fig 3D and E). Numerous RAD51 foci were also detected in *rad51_AtRAD51-II3A* plants, demonstrating the ability of the RAD51-II3A protein to bind DNA in vivo and assemble at DNA breaks (Fig 3A–C). Interestingly, we also observed RAD51 foci in untreated *rad51_AtRAD51-II3A* plants (Fig 3A), with 30% of the nuclei exhibiting 1–2 or 3–10 RAD51 foci (Fig 3D and E), suggesting the

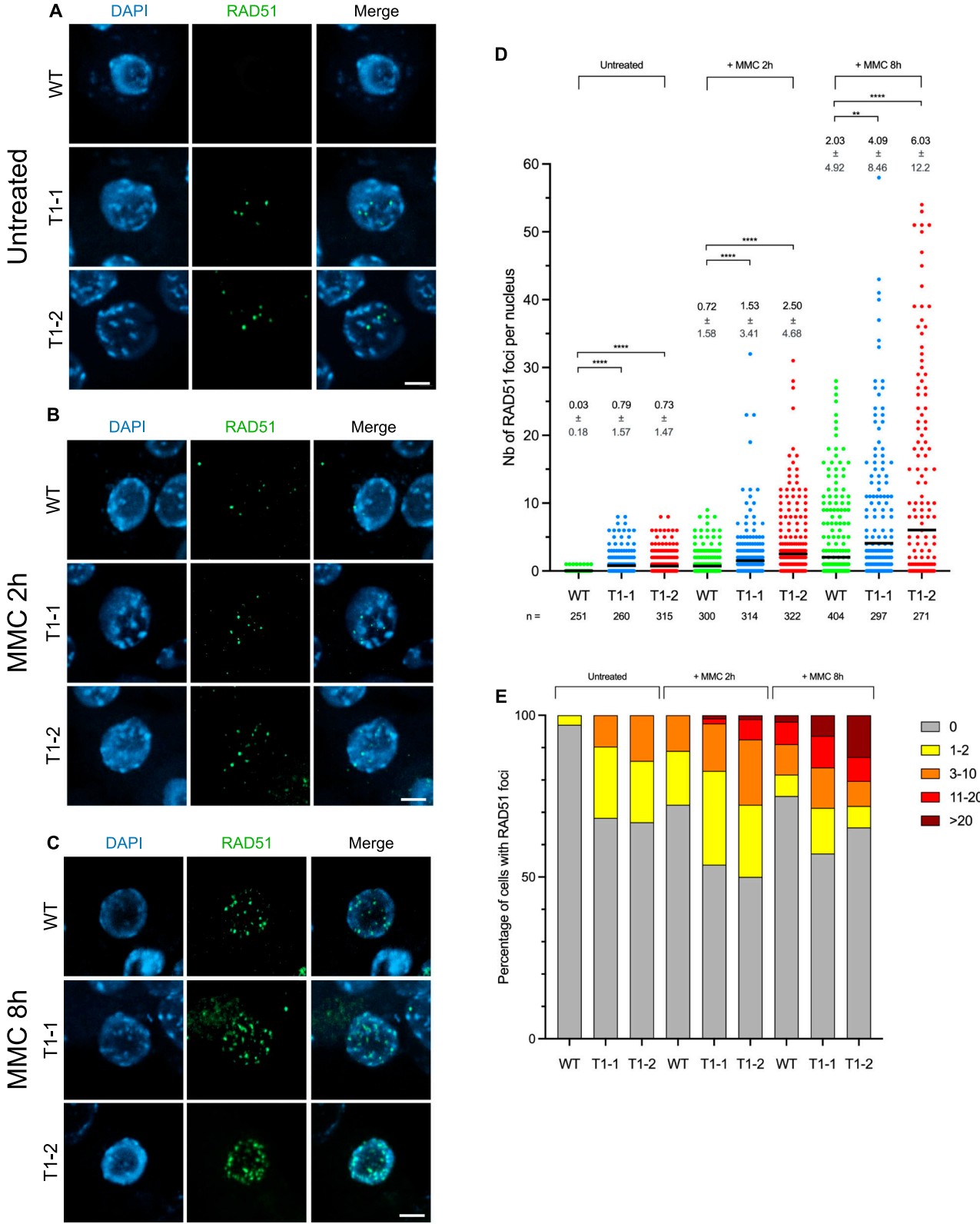

**Figure 3. AtRAD51-II3A focus formation in somatic cells.**
**(A, B, C)** Immunolocalization of RAD51 in root tip nuclei of untreated seedlings (A), or seedlings 2 h (B) or 8 h (C) after treatment with 30 $\mu$M mitomycin C (MMC). Experiments were conducted on two independent transgenic *rad51* lines carrying RAD51-II3A (named T1-1 and T1-2). Scale bars: 5 $\mu$m. **(D)** Number of RAD51 foci per nucleus in transgenic *rad51*_RAD51-II3A lines compared with WT before or 2 and 8 h after treatment with 30 $\mu$M MMC. Data are presented as mean ± SD. n indicates number of cells

presence of unrepaired DSBs and/or nonspecific binding of AtRAD51-II3A. In accordance, both rad51_AtRAD51-II3A lines showed an increased number of DNA damage–induced RAD51 foci after MMC treatment compared with WT plants (see, e.g., fraction of cells with more than 20 foci; Fig 3D and E). No major difference is observed between the two rad51_AtRAD51-II3A lines. We next quantified the fluorescence intensity of RAD51 foci to assess whether or not the binding of AtRAD51-II3A is affected (Fig S1). Interestingly, we observed a significantly higher mean and sum intensities per RAD51 focus for both rad51_AtRAD51-II3A lines compared with RAD51 foci in WT plants (Fig S1). Thus, AtRAD51-II3A binding is not decreased, rather there appears to be an increased accumulation of RAD51 at DNA break sites in the rad51_AtRAD51-II3A lines, possibly as a result of delayed repair of the DNA breaks. The AtRAD51-II3A protein thus retains the ability to bind DNA in vivo and assemble at DNA break sites.

### AtRAD51-II3A is severely defective in DNA repair and HR in vivo

Next, we wanted to test whether AtRAD51-II3A shows separation of function activity in vivo (i.e., separation of RAD51's DNA-binding and recombinogenic activity) as previously shown for S. cerevisiae and human RAD51-II3A (Cloud et al, 2012; Mason et al, 2019). We thus tested sensitivity of rad51_AtRAD51-II3A plants to the DNA damaging agent MMC. Plants were grown on solid media supplemented, or not, with 30 µM MMC and sensitivity was scored after 2 wk. Under standard conditions, rad51_AtRAD51-II3A plants did not exhibit any visible phenotypical difference with respect to the WT (Fig 4A and B). In contrast, rad51_AtRAD51-II3A plants display a strong hypersensitivity to MMC (Fig 4A and B). As control, we also tested rad51 mutant plants expressing a WT RAD51 (named RAD51g for RAD51 genomic sequence, non-mutated) and, as anticipated, rad51_RAD51g plants do not exhibit MMC hypersensitivity (Fig 4A and B). This result suggests that AtRAD51-II3A has lost its recombinogenic activity. Interestingly, we also observed strong MMC hypersensitivity of RAD51+/−_AtRAD51-II3A (plants heterozygous for endogenous RAD51 carrying the AtRAD51-II3A transgene) and RAD51+/+_AtRAD51-II3A plants even in presence of endogenous RAD51 (Fig 4A and B, RAD51+/−_AtRAD51-II3A plants and Fig S2). Thus, AtRAD51-II3A acts as a dominant negative and disturbs the function of the native RAD51. This is reminiscent of the phenotype of plants expressing RAD51-GFP fusion protein, but in striking contrast with both S. cerevisiae Rad51-II3A and Rad51-GFP (Cloud et al, 2012; Da Ines et al, 2013; Waterman et al, 2019).

To extend this analysis, we directly tested somatic homologous recombination using the well-characterized IU.GUS recombination tester locus, specifically measuring RAD51-dependent recombination (Orel et al, 2003; Roth et al, 2012). This tester locus consists of an interrupted, nonfunctional β-glucuronidase (GUS) gene and a template GUS sequence for repair. Productive GUS recombination at the IU.GUS locus using this internal template GUS sequence restores the complete functional GUS gene which is

scored histochemically as blue tissue sectors. The IU.GUS recombination reporter locus was inserted into rad51_AtRAD51-II3A T1-1 plants and somatic HR frequencies monitored in progeny homozygous for IU.GUS (Fig 4C and Table 1). WT plants have a mean of 2.5 recombination events per plant (SEM = 0.27; n = 186). Remarkably, HR is dramatically altered in rad51_AtRAD51-II3A plants (no recombinant spot found in n = 176 plants). Given the dominant negative MMC-hypersensitivity, we also tested somatic HR in WT plants expressing AtRAD51-II3A. Somatic HR is also severely reduced in these plants with a mean of 0.6 recombination events per plant (SEM = 0.14; n = 116; Fig 4C and Table 1), in accordance with the dominant negative phenotype conferred by AtRAD51-II3A. This confirms that AtRAD51-II3A has a separation of function phenotype, retaining DNA-binding activity but being defective for homologous recombination and repair.

### AtRAD51-II3A mutation does not impair meiotic recombination

Previous data have demonstrated that RAD51 recombinogenic activity is not essential for meiotic DSB repair. This has been demonstrated directly in S. cerevisiae using the ScRad51-II3A separation-of-function mutant (Cloud et al, 2012) and indirectly in Arabidopsis through the characterization of RAD51-GFP plants (Da Ines et al, 2013). As anticipated, AtRAD51-II3A restores fertility of Arabidopsis rad51 mutant plants (Fig 5A) and AtRAD51-II3A focus formation appears normal in meiotic cells (Fig 5B). Neither is meiotic progression affected by the presence of AtRAD51-II3A in rad51 mutant plants, as confirmed by cytogenetic analyses of male meiosis stages (Fig 5C). Arabidopsis rad51 mutants exhibit synapsis defects at late prophase I and massive chromosome fragmentation largely visible at later stages (Fig 5C) (Li et al, 2004). In contrast, and alike WT plants, rad51_AtRAD51-II3A plants showed synapsed chromosomes at late prophase I, five aligned bivalents at metaphase I that segregate normally at anaphase I. Eventually, four balanced nuclei are observed at the end of meiosis (Fig 5C). Importantly, we exclusively observed bivalents in all Metaphases I analyzed (n = 10) and we did not observe meiotic defects in our cytological analyses.

### RAD51 site II function is not essential for meiotic recombination when DMC1 is present

AtRAD51-II3A fully complements rad51 meiotic defects, notwithstanding its defective recombinogenic activity. To demonstrate that this is due to the fact the recombinogenic activity of RAD51 is not essential for meiotic recombination as this is supplied by DMC1, we introduced the AtRAD51-II3A allele into a dmc1 background. In the Arabidopsis dmc1 mutant, meiotic DSBs are repaired by RAD51, presumably using the sister chromatid, and intact univalents (no bivalents) are observed at metaphase I (Fig 6) (Couteau et al, 1999; Pradillo et al, 2012). These randomly segregate at anaphase I and abnormal

---

analyzed. Kruskal–Wallis test; **P-value < 0.01, ****P-value < 0.0001. **(E)** Percentage of cells with 0, 1–2, 3–10, 11–20, and >20 RAD51 foci is shown for each genotype without or after MMC treatment. **(D)** Number of cells analyzed is the same as in (D).
Source data are available for this figure.

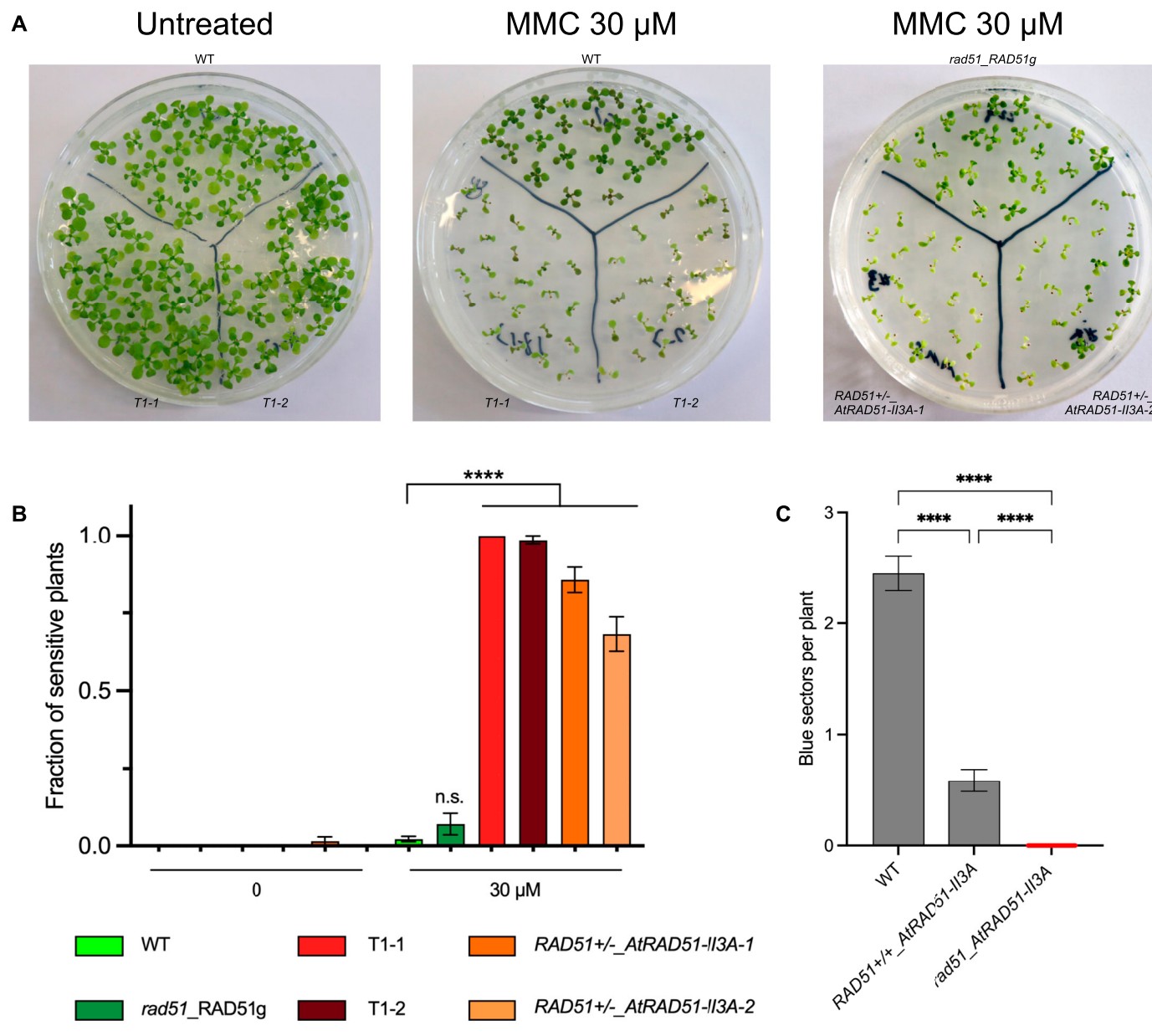

**Figure 4. AtRAD51-II3A is defective in double-strand break repair and HR in somatic cells.**
**(A)** Pictures of 2-wk-old seedlings grown without (left) or with (middle and right pictures) 30 μM mitomycin C. **(B)** Fraction of sensitive plants was estimated based on the number of true leaves per seedling. Seedlings with three or less true leaves were considered sensitive to DNA damage. Bars are mean ± SEM of at least three independent experiments (except for *rad51_RAD51g* for which two replicates were performed) with 20–24 seedlings per genotype per experiment. Error bar for T1-1 is not visible because all replicates exhibited 100% of sensitive plants. Two-way ANOVA test; ****$P$-value < 0.0001, n.s. not significant. **(C)** Quantification of spontaneous somatic HR events using the IU.GUS reporter system. HR events were quantified as the number of blue spots per seedling. 52–62 seedlings were analyzed per genotype, and two to three biological replicates were performed per genotype. Bars indicate mean ± SEM. Kruskal–Wallis test; ****$P$-value < 0.0001.
Source data are available for this figure.

polyads are observed at the end of meiosis (Fig 6). We observed severe chromosome fragmentation in *dmc1_rad51*_RAD51-II3A plants, unambiguously demonstrating that AtRAD51-II3A lacks recombinogenic activity. We also note that, as previously observed for somatic recombination, AtRAD51-II3A has a dominant negative effect and extensive chromosome fragmentation is also observed in presence of endogenous RAD51 (Fig 6, *dmc1*_RAD51+/−_*AtRAD51-II3A* and *dmc1*_RAD51+/+_*AtRAD51-II3A*).

## Discussion

We have built and characterized the Arabidopsis AtRAD51-II3A separation-of-function mutant and show that RAD51 DNA-binding site II function is required for RAD51 recombinogenic activity in Arabidopsis, but not nucleofilament formation. These conclusions concord with previous data in *S. cerevisiae* and human (Cloud et al, 2012; Mason et al, 2019; Ito et al, 2020) and confirm the conservation of

**Table 1.  Spontaneous somatic homologous recombination events in WT and AtRAD51-II3A plants.**

|  | n | N | m ± SEM | P |
|---|---|---|---|---|
| WT | 62 | 154 | 2.48 ± 0.254 |  |
|  | 62 | 166 | 2.68 ± 0.3 |  |
|  | 62 | 135 | 2.18 ± 0.244 |  |
| *RAD51+/+_AtRAD51-II3A* | 54 | 34 | 0.63 ± 0.155 | <0.0001[a] |
|  | 62 | 34 | 0.55 ± 0.133 |  |
| *rad51_AtRAD51-II3A* | 62 | 0 | 0 ± 0 | <0.0001[b] |
|  | 52 | 0 | 0 ± 0 |  |
|  | 62 | 0 | 0 ± 0 | <0.0001[c] |

n indicates the number of seedlings analyzed; N, total number of blue spots (recombination events); m ± SEM, mean number of recombination events per plant. *P* was calculated using nonparametric statistical analysis (Kruskal–Wallis test).
[a]Indicate statistical analysis between WT and *RAD51+/+_AtRAD51-II3A*.
[b]Indicate statistical analysis between WT and *rad51_AtRAD51-II3A*.
[c]Indicate statistical analysis between *RAD51+/+_AtRAD51-II3A* and *rad51_AtRAD51-II3A*.

site II function in plants. Structural features of DNA-binding sites in RecA-like family recombinases seem thus to be highly conserved. The *AtRAD51-II3A* separation-of-function mutant includes three substitution mutations, one of which is located in Walker A motif (R133 in Arabidopsis). The respective contribution of each of these amino acids in the RAD51 strand exchange activity is unclear. The Walker A motif plays a key role in the binding and hydrolysis of ATP. Consequently, mutation of R133 in Walker A might well affect ATP binding and/or hydrolysis, which could in turn stabilize the RAD51 presynaptic filament, preventing it from subsequent strand exchange activity. However, a recent study using *Schizosaccharomyces pombe* Rad51 elegantly showed that mutation of only the two site II amino acids R324 and K334 (equivalent to Arabidopsis R306 and K316) leads to severe loss of strand exchange activity, whereas the mutant protein exhibits WT level of ATP hydrolysis (Ito et al, 2020). These data thus strongly argue that loss of strand exchange activity in site II mutants cannot be solely explained by defect in binding or hydrolysis of ATP.

### DNA-binding site II is required for AtRAD51 catalytic activity

RAD51 is essential for both mitotic and meiotic DNA DSB repair. It binds ssDNA to form a nucleofilament that searches for and invades a second DNA molecule to use it as a repair template. This two-step process requires distinct parts of the RAD51 protein: site II is not involved in ssDNA binding and nucleofilament formation but becomes essential for further steps of the recombination process. Confirming this, we show that Arabidopsis RAD51-II3A retains DNA-binding ability in vivo by immunolocalizing RAD51 in both somatic and meiotic cells. We further demonstrate that AtRAD51-II3A is unable to repair DNA DSB in somatic and meiotic cells, showing that it has lost its recombinogenic activity. This is reminiscent of *S. cerevisiae* and human RAD51-II3A and RAD51-GFP in Arabidopsis (Cloud et al, 2012; Da Ines et al, 2013; Mason et al, 2019). Our data also highlight a dominant negative effect of the *AtRAD51-II3A* separation-of-function mutant, analogous to that observed with

the RAD51-GFP in Arabidopsis (Da Ines et al, 2013), and human RAD51-II3A (Saayman et al, 2023). This is strikingly different from both *S. cerevisiae* Rad51-II3A and Rad51-GFP, neither of which exhibit a dominant negative effect (Cloud et al, 2012; Waterman et al, 2019). We hypothesize that this dominant negative effect is the result of a greater affinity to ssDNA. Indeed, *S. cerevisiae* Rad51-II3A has a small DNA-binding defect relative to WT Rad51 and makes fainter foci (Cloud et al, 2012; Ito et al, 2020). This DNA-binding defect is not seen for human RAD51 (Mason et al, 2019) and our data on RAD51 foci intensity suggest it is also not the case for Arabidopsis RAD51-II3A. Thus, whereas *S. cerevisiae* Rad51-II3A may not strongly compete with endogenous Rad51 for making filament, both human and Arabidopsis RAD51-II3A may compete more effectively, impeding binding of endogenous RAD51. The mixed RAD51/RAD51-II3A nucleofilament thus formed would likely not be productive for recombination. This hypothesized mechanism, however, remains to be tested directly.

### Activity of RAD51 DNA-binding site II is not required for meiotic DSB repair in Arabidopsis

Previous studies have demonstrated that RAD51 plays a supporting role for DMC1 recombinogenic activity during meiosis (Cloud et al, 2012; Da Ines et al, 2013; Liu et al, 2014; Hinch et al, 2020; Chen et al, 2021). In plants, this has been shown indirectly using an RAD51-GFP fusion protein lacking D-loop invasion activity (Da Ines et al, 2013; Kobayashi et al, 2014). Here, we confirm that RAD51 DNA-binding site II recombinogenic activity is not required for meiotic break repair, incidentally confirming that DMC1 is the main active recombinase during meiotic recombination. We also clearly demonstrate that the activity carried by RAD51 DNA-binding site II is not essential for meiosis. It is now accepted that RAD51 recombinogenic activity is inhibited during meiosis, and although this is starting to be well understood in *S. cerevisiae* (Brown & Bishop, 2014; Callender et al, 2016; Emmenecker et al, 2023), it remains little characterized in multicellular eukaryotes. Recent evidence from our laboratory suggests that DMC1 might be directly involved in RAD51 down-regulation (Da Ines et al, 2022) and the data presented here suggest that this inhibition may occur through down-regulation of DNA-binding site II function.

## Materials and Methods

### Plant material and growth conditions

The following mutant plant lines were used: *rad51-1* (Col-0 background; [Li et al, 2004]), *dmc1-2* (Col-0 background; [Pradillo et al, 2012]), and IU-GUS.8 (Col-0 background; [Orel et al, 2003]). *A. thaliana* seeds were stratified at 4°C for 2 d and grown on soil or in vitro (on 0.5X Murashige and Skoog salts medium [M0255; Duchefa Biochemie] with 1% sucrose, 0.8% [m/v] agar) in a growth chamber under a 16:8 h light:dark photoperiod at 23°C with 60% relative humidity. For in vitro culture, seeds were first surface sterilized in 70% ethanol/0.05% SDS for 5 min, washed in 95% ethanol, and air-dried. For selection of primary transformants, hygromycin (15 μg/ml) and cefotaxin (100 μg/ml) were added to the medium, stratified for 2 d at 4°C, and then grown at 23°C as described above.

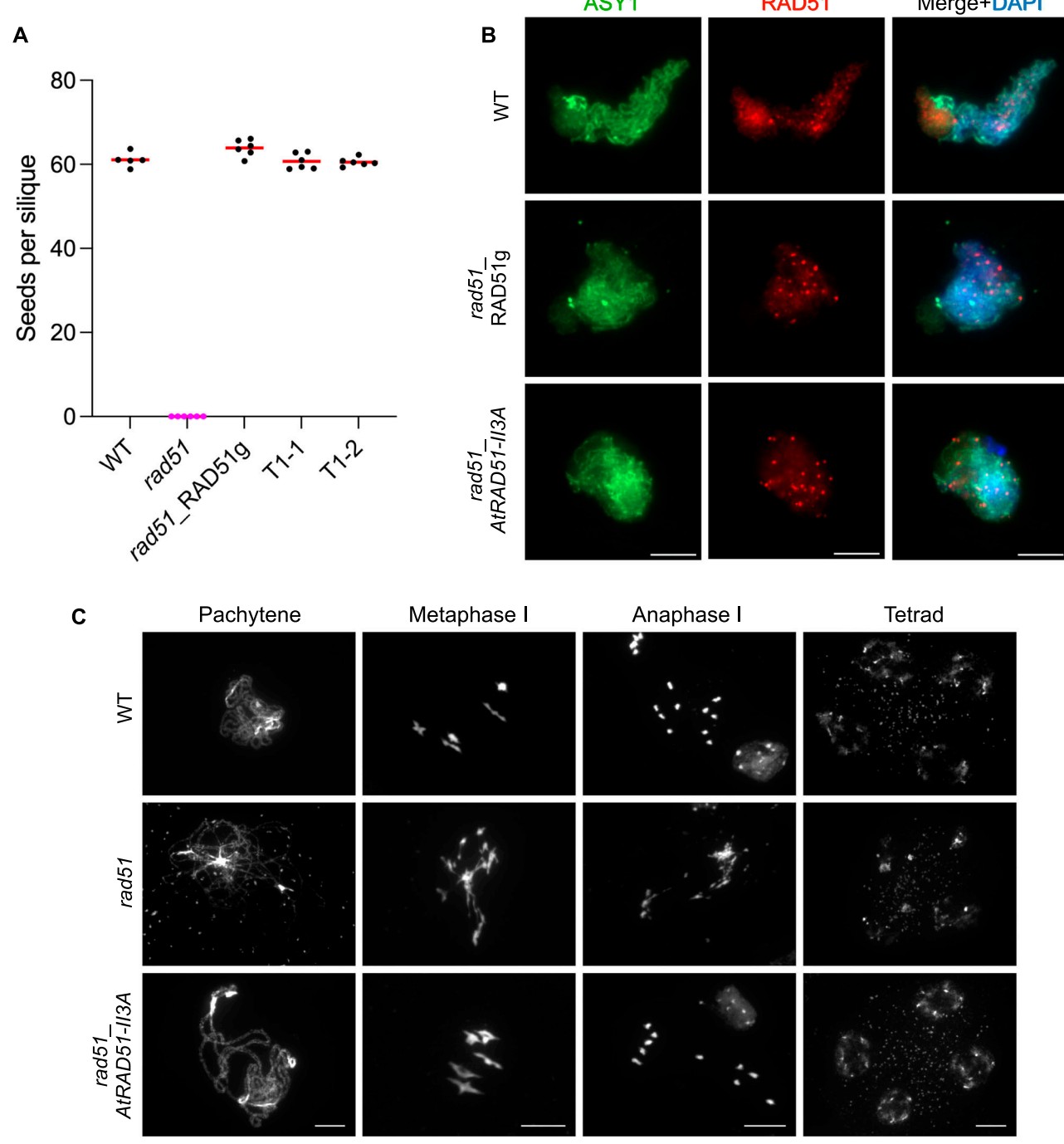

**Figure 5. AtRAD51-II3A restores fertility and meiotic progression of *rad51* mutant.**
**(A)** Comparison of fertility based on the number of seeds per silique in WT, *rad51*, *rad51* complemented with RAD51 genomic sequence (RAD51g) and *rad51_AtRAD51-II3A* lines. Each dot represents the mean number of seeds per silique for one plant, obtained by counting at least 12 siliques. Five to six plants were analyzed per genotype. No significant difference was measured between WT, *rad51*_RAD51g, *rad51_AtRAD51-II3A* T1-1, and T1-2. Kruskal–Wallis test. **(B)** RAD51/ASY1 co-immunolocalization on late prophase I–staged meiocytes. Scale bars: 5 μm. **(C)** DAPI-stained chromosome spread of male meiocytes at late prophase I, metaphase I, anaphase I, and tetrad stage. Scale bars for each stage: 10 μm.
Source data are available for this figure.

### Cloning of RAD51-II3A and plant transformation

The complete genomic region of RAD51 from ATG to stop codon was synthesized (Integrated DNA Technologies Inc.) with mutations to convert R133, R306, and K316 into alanine (Fig 1). The synthesized RAD51-II3A was then cloned into the GATEWAY destination vector pMDC32 in which the 35S promoter was replaced by the RAD51 promoter (1,031 bp 5′ upstream sequence of RAD51; [Da Ines et al,

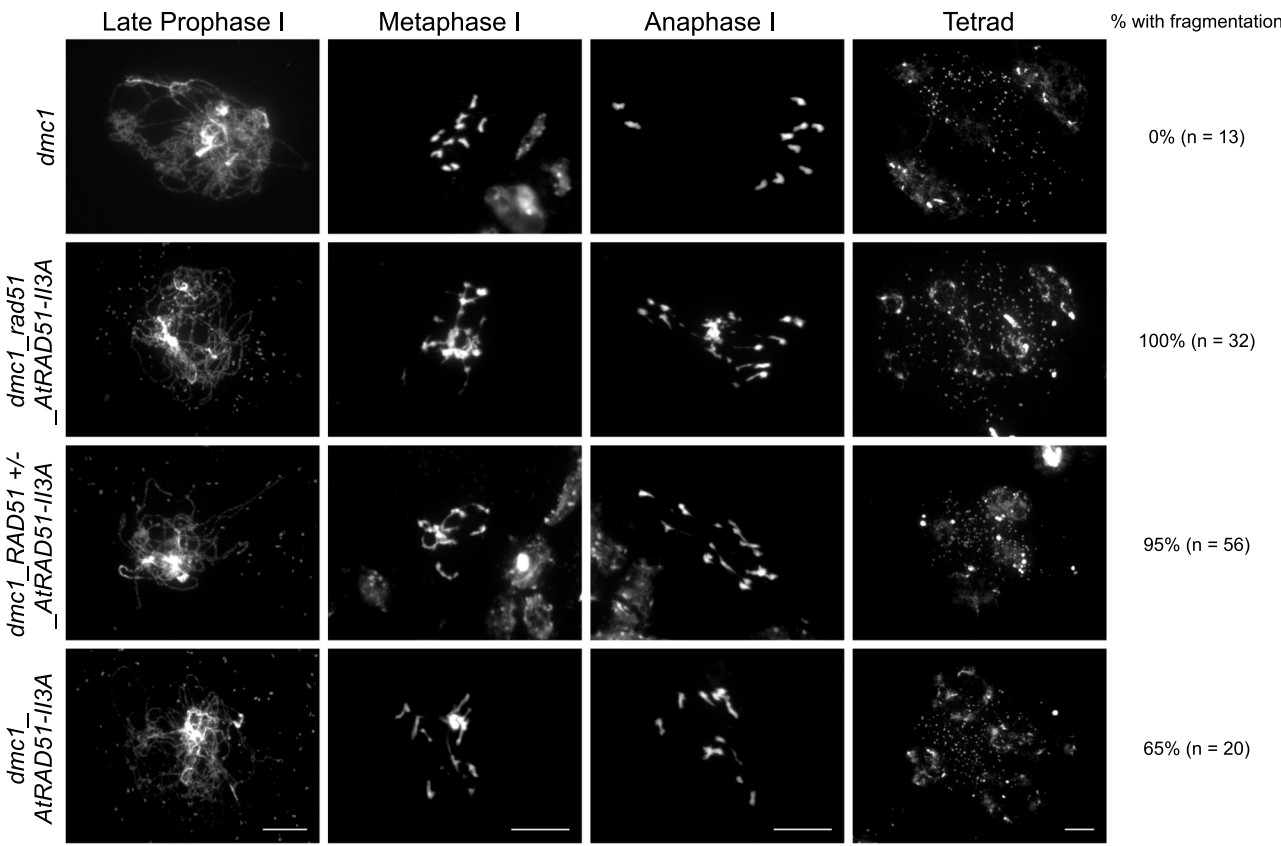

**Figure 6. AtRAD51-II3A does not repair meiotic double-strand break.**
DAPI-stained chromosome spread of male meiocytes at late prophase I, metaphase I, anaphase I, and tetrad stage. Plants expressing AtRAD51-II3A show massive chromosome fragmentation. Percentage of cells showing chromosome fragmentation and number of cells analyzed is indicated on the right. Scale bars for each stage: 10 μm.

2022]). The plasmid was then inserted in an *Agrobacterium tumefaciens* C58C1 strain which was used to transform Arabidopsis plants by floral dip according to Clough and Bent (1998). Seeds from the transformed plants were harvested and selected in vitro for hygromycin resistance.

### Protein structure prediction and comparison

We used the AlphaFold structure prediction of Arabidopsis RAD51 (AF-P94102-F1 model) and the published pdb (protein data bank format) for *Saccharomyces cerevisiae* Rad51 (PDB ID:1szp) (Conway et al, 2004) and human RAD51 (PDB ID:5h1c) (Xu et al, 2017). All structures were visualized and superimposed using ChimeraX (Pettersen et al, 2021).

### MMC sensitivity assay

Seeds were surface sterilized and sown onto solid medium (0.5X MS, 0.8% [m/v] agar, and 1% sucrose), supplemented or not with 30 μM MMC (Sigma-Aldrich). Seeds were stratified for 2 d at 4°C and further grown at 23°C for 2 wk. MMC sensitivity was measured by counting the number of true leaves as previously described (Bleuyard & White, 2004). Plants with less than four true leaves were considered sensitive.

Statistical analysis was performed with Two-way ANOVA test (GraphPad Prism v10.1.1 software).

### Histochemical GUS staining for somatic homologous recombination assay

The frequency of somatic homologous recombination was determined by using the IU-GUS.8 line containing an interrupted *ß-glucuronidase* (*GUS*) gene (Orel et al, 2003).

Seeds were surface sterilized, stratified at 4°C for 2 d, and grown in vitro on MS medium for 2 wk. Seedlings were then incubated in GUS staining buffer (0.2% Triton X-100, 50 mM sodium phosphate buffer, pH 7.2, and 2 mM X-Gluc [Biosynth] dissolved in N,N-dimethylformamide). Seedlings were vacuum-infiltrated for 15 min and incubated at 37°C for 24 h. Staining solution was then replaced with 70% EtOH to remove leaves pigmentation and blue spots were counted under a binocular microscope. Statistical analysis was performed with Mann-Whitney test (GraphPad Prism v10.1.1 software).

### Fertility analysis

Between 25 and 30 siliques from the primary stem were collected and bleached in a 95% Ethanol bath at 70°C for several hours. The

number of seeds per silique was then counted manually under a binocular microscope. All analyzed plants were grown side-by-side. Statistical analysis was performed with Kruskal–Wallis test (GraphPad Prism v10.1.1 software).

### RAD51 immunolocalization on root nuclei

Seedlings were grown for 5 d on MS medium and fixed in 4% PFA-1X PME for 45 min. Immunostaining in root tip nuclei was then performed as previously described (Charbonnel et al, 2010). Slides were incubated with rat $\alpha$-RAD51 (diluted 1/500 in 3% BSA, 0.05% Tween-20 in 1X PBS) in a moist chamber at 4°C overnight. Slides were washed three times in 1X PBS-0.05% Tween-20, air-dried, and then incubated with secondary antibody solution (chicken anti-rat Alexa 488 [Invitrogen] diluted 1/1,000 in 3% BSA, 0.05% Tween-20 in 1X PBS) in a moist chamber for 3 h at RT in the dark. Slides were finally washed three times in 1X PBS-0.05% Tween-20, air-dried, and mounted in VECTASHIELD mounting medium containing DAPI (1.5 $\mu$g/ml; Vector Laboratories Inc.).

Z-stacks images were acquired with a Zeiss Cell Observer Spinning Disk microscope and analyzed using Imaris software v9.8.2. 3D root nuclei were segmented using the segmentation tool. A mask was created on these segmented surfaces to display EGFP (Channel 2) only in the surfaces (Channel 3). A random color mask was then applied on the DAPI channel (Channel 1) to give each surface a unique color ID (Channel 4). Eventually, spots were created with the "Spots" tool on RAD51 foci using Sum Square of channel 3 as a quality control. Statistics (Surface: Volume, Median Intensity of Ch4; Spots: Intensity Min, Max, Mean, Median, Sum, SD, Sum Square of Ch2, Median Intensity Ch4) were exported and data plotted using GraphPad Prism v10.1.1. Statistical analysis of the number of RAD51 foci per nucleus was performed using Kruskal–Wallis test (GraphPad Prism v10.1.1 software).

### Meiotic chromosome spreads

Meiotic chromosome spreads were prepared as described in Ross et al (1996). Inflorescences collected from secondary stems were fixed in Carnoy's fixative (3:1 ethanol:acetic acid), and washed once in ultrapure water then twice in 10 mM citrate buffer (pH 4.5). Flower buds were then digested in enzyme mixture (0.3% cellulase, 0.3% pectolyase, and 0.3% cyclohelicase; Sigma-Aldrich) for 3 h at 37°C in a moist chamber. Reaction was stopped by placing the slides on ice and replacing enzyme mix with ice-cold 10 mM citrate buffer (pH 4.5). Immature flower buds of appropriate stage (0-3-0.6 mm) were then selected under a binocular microscope, placed individually on a clean microscope slide, and crushed with a dissection needle. Chromosomes were spread by stirring for 1 min in 20 $\mu$l 60% acetic acid at 45°C, fixed with Carnoy's fixative, and air-dried. Finally, slides were mounted in VECTASHIELD mounting medium containing DAPI (1.5 $\mu$g/ml DAPI; Vector Laboratories Inc.) and covered with 24 × 32-mm coverslip. Images were acquired with a Zeiss AxioImager.Z1 epifluorescence microscope equipped with an Axio-Cam Mrm camera and DAPI filter.

### Immunolocalization of meiotic proteins in pollen mother cells

Spreads of pollen mother cells for immunolocalization of RAD51 were prepared as described previously (Armstrong et al, 2002). Primary antibodies used for immunostaining were anti-ASY1 raised in guinea Pig (1:500; Armstrong et al, 2002) and anti-RAD51 raised in rats (1:100; [Kurzbauer et al, 2012]). Secondary antibody: anti-rat Alexa Fluor 488 and anti-guinea pig Alexa Fluor 594 were used at 1:100 dilution. Images were obtained using a Zeiss AxioImager.Z1 epifluorescence microscope equipped with an Axio-Cam Mrm camera and were analyzed using Zeiss Zen Lite software.

## Supplementary Information

## Acknowledgements

We thank Peter Schlögelhofer and Eugenio Sanchez-Moran for providing the RAD51 and ASY1 antibodies, respectively. We thank members of the recombination group for their help and discussions. This work was supported by the CNRS, INSERM, Université Clermont Auvergne, ANR-16-CE91-0010-01 RecInChromatin (to CI White) and the European Union (H2020-MSCA-ITN2017: 765212-MEICOM to CI White). The funders had no role in study design, data collection and analysis, decision to publish, or preparation of the manuscript.

### Author Contributions

V Petiot: conceptualization, data curation, formal analysis, validation, investigation, visualization, methodology, and writing—original draft, review, and editing.
CI White: conceptualization, supervision, funding acquisition, validation, project administration, and writing—review and editing.
O Da Ines: conceptualization, data curation, formal analysis, supervision, funding acquisition, validation, investigation, visualization, methodology, project administration, and writing—original draft, review, and editing.

### Conflict of Interest Statement

The authors declare that they have no conflict of interest.

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
