## [Reviewer comments · Life Science Alliance]

Life Science Alliance

DNA binding site II is required for RAD51 recombinogenic activity in *Arabidopsis thaliana*

Valentine PETIOT, Charles White, and Olivier Da Ines

DOI: <https://doi.org/10.26508/lsa.202402701>

Corresponding author(s): *Olivier Da Ines, Genetique Reproduction and Developpement*

Review Timeline:

Submission Date:	2024-03-07
Editorial Decision:	2024-04-05
Revision Received:	2024-04-29
Editorial Decision:	2024-05-02
Revision Received:	2024-05-03
Accepted:	2024-05-07

Scientific Editor: *Eric Sawey, PhD*

Transaction Report:

April 5, 2024

Re: Life Science Alliance manuscript #LSA-2024-02701

Dr. Olivier Da Ines
Genetique Reproduction and Developpement
CNRS UMR6293
28 Place Henri Dunant
Clermont-Ferrand 63001
France

Dear Dr. Da Ines,

Thank you for submitting your manuscript entitled "DNA binding site II is required for RAD51 recombinogenic activity in *Arabidopsis thaliana*" to Life Science Alliance. The manuscript was assessed by expert reviewers, whose comments are appended to this letter. We invite you to submit a revised manuscript addressing the Reviewer comments.

Thank you for this interesting contribution to Life Science Alliance. We are looking forward to receiving your revised manuscript.

Sincerely,

B. MANUSCRIPT ORGANIZATION AND FORMATTING:

Reviewer #1 (Comments to the Authors (Required)):

To characterize the importance of RAD51 DNA binding site II, in particular for homologous recombination in somatic and meiotic cells in Arabidopsis, Petiot et al. generated rad51_RAD51-II3A plants and performed phenotypic analysis, immunostaining, GUS staining and chromosome spread experiments to conclude that "DNA binding site II function is required for RAD51 recombinogenic activity in Arabidopsis, but not nucleofilament formation." In addition, the authors made use of RAD51+/-_AtRAD51-II3A plants to show AtRAD51-II3A has a dominant negative effect, differing from the conclusion in budding yeast but similar to that in human.

The paper provides many interesting and important details to characterize the functions of RAD51 DNA binding site II in DNA damage repair in Arabidopsis although this is not novel in yeast and human study.

Major comments:

1. The references to organisms in this manuscript are not consistent. E.g. ".....Arabidopsis RAD51 based on the yeast and human Rad51-II3A....." (Line 141); ".....of *S. cerevisiae* and human RAD51 (Figure 1)." (Line 143); ".....with budding yeast and human RAD51," (Line 144). The references to organisms can be either Latin or common names but should be consistent throughout the manuscript. Please check the references throughout the whole manuscript.

2. From the provided data (Fig1), Arg133 (R133) is in Walker A not in Site II, could the authors explain it?

3. IU-GUS.8 line is used for somatic homologous recombination assay; blue spots indicate the recombination events. There is no single spot found in 176 rad51_RAD51-II3A plants. I was wondering whether the authors checked the rad51_RAD51-II3A plants carrying the IU-GUS.8 or not. If so, please provide the data.

Minor points:

4. Line 252: ".....5 aligned bivalents at metaphase.....", first letter should be capitalized for metaphase.

5. Figure 2: The positions and names of three essential amino acids should be written next to them.

6. Figure 4B: Error bar is missing.

7. Figure 5B: Scale bar is missing; 5C: the photos for Anaphase I in WT and rad51_RAD51-II3A are in Telophase I. Please select the correct photos. And remove "10 µm" from the photos.

8. Figure 6: Remove "10 µm" from the photos.

9. Figure S2: In S2A it is written as "rad51_RAD51g" but in S2B it is "rad51+RAD51g", please keep the labels consistent; the error bars are missing S2B.

Reviewer #2 (Comments to the Authors (Required)):

Previous studies in yeasts and humans have shown that the DNA binding Site II of RAD51 is essential for somatic recombination but not for meiotic recombination. In addition, RAD51 with the mutation in DNA binding Site II (RAD51-II3A) showed a dominant negative effect in humans. Basically, this study showed that RAD51-II3A has the same effect in Arabidopsis. The data supported the conclusion well.

(1) I have some difficulties with the genotype of transgenic lines. "AtRAD51-II3A", "RAD51-II3A" and "rad51_AtRAD51-II3A" were used. Are they the same? If so, please be consistent. What was the genotype of RAD51 in "RAD51+/-_AtRAD51-II3A"? How the genotype was determined?

(2) In Fig. 2: The resolution of the lower panel of Fig.2A needs to be improved. In addition, the Close-up views in Fig. 2B and 2C were not consistent with the views in the square.

(3) In Fig. 4B: Please perform statistical analysis.

(4) In Fig. S1: Compared with the untreated condition, why the mean intensity and sum intensity of RAD51 foci in T1-1 were lower after MMC treatment. Should they be higher?

Reviewer #3 (Comments to the Authors (Required)):

The manuscript by Valentine et al. through analysis of Arabidopsis RAD51-II3A separation-of-function mutant showed that the RAD51 DNA-binding site II function is required for the recombination activity, but not for nuclear filament formation. Furthermore, the authors demonstrated that the activity of RAD51 DNA binding site II is unnecessary for meiosis. These conclusions are consistent with previous findings from both yeast and humans, which support the idea that the Site II function is conserved across species. The manuscript is well organized and presented in a clear manner. The finding is pretty novel in plants. I have a few concerns that should be considered by the authors to further improve the manuscript.

1. As described in the manuscript, R133 is located in the Walker A domain, while R306A and K316A are located in the DNA binding site II. Because only one of the three mutated amino acids is located in Walker A, as named DNA binding site II is confusing. The conserved R133 in Walker A is thought to affect the binding and hydrolysis of ATP, is it possible that its mutation may prevent ATP hydrolysis, thus stabilizing the RAD51 presynaptic filament from entering subsequent catalysis. Given the fact that AtRAD51-II3A has retained the DNA binding activity and is defective of homologous recombination and repair, which lead to a separation of function phenotype. As a result, this study highlights that DNA binding site II is essential for RAD51 recombinogenic activity. However, the potential function of the other two sites (R306A and K316A) should be at least discussed.
2. As described from Line 150 to 156, however, the hDMC1 sequence used by Xu et al, 2017 is Q313 rather than K313, and the residue in the structure (PDB 5h1c) is also Q, so is Q visible in the structural superposition in Fig2C.
3. Line 207 on pg9 , "MMC induces DNA interstrand crosslink adducts and in turn, DNA strand breaks that are repaired by RAD51-dependent homologous recombination. " The author should explain MMC function when it first appears.
4. Line 241 on pg10 , "RAD51-II3A mutation does not impair meiotic recombination." delete the period, because the other headings do not have a period; Line 333 on pg14, 15µg/ml with less space in the middle; Line 376 on pg15, "Immunolocalization" The first letter "I" does not need to be capitalized ; Lines 607 and 610 on pg23, 30µg/ml, with less space in the middle; "or seedlings 2h (B) or 8h (C)" B should be bold.
5. Fig1B legend , Walker domains are outlined in magenta instead of pink.
6. In Fig 2B-C, three residues should be marked. The color of the residues corresponding to hRAD51/ScRAD51 should also be described. In Fig5B, Bar is not obvious. The data at the bottom of Fig2A is neither mentioned in the text nor the legend.
7. The "Homologous" in the title of Table1 should be lowercase.

Dear Editor,

We're very happy with the positive reactions of all three reviewers to our manuscript and thank them for their suggestions. We have now responded to all reviewers' comments. Our responses to the points raised by all three reviewers and the changes we've made in the text are detailed below (The reviewers' texts are in blue 11-point type and our text in black 11-point type).

We now hope that our manuscript will be suitable for publication in Life Science Alliance.

Sincerely,

Reviewer #1 (Comments to the Authors (Required)):

To characterize the importance of RAD51 DNA binding site II, in particular for homologous recombination in somatic and meiotic cells in Arabidopsis, Petiot et al. generated *rad51_RAD51-II3A* plants and performed phenotypic analysis, immunostaining, GUS staining and chromosome spread experiments to conclude that "DNA binding site II function is required for RAD51 recombinogenic activity in Arabidopsis, but not nucleofilament formation." In addition, the authors made use of *RAD51+/-_AtRAD51-II3A* plants to show *AtRAD51-II3A* has a dominant negative effect, differing from the conclusion in budding yeast but similar to that in human.

The paper provides many interesting and important details to characterize the functions of RAD51 DNA binding site II in DNA damage repair in Arabidopsis although this is not novel in yeast and human study.

We thank the reviewer for her/his positive comments on our manuscript.

Major comments:

1. The references to organisms in this manuscript are not consistent. E.g. ".....Arabidopsis RAD51 based on the yeast and human Rad51-II3A....." (Line 141); ".....of *S. cerevisiae* and human RAD51 (Figure 1)." (Line 143); ".....with budding yeast and human RAD51," (Line 144). The references to organisms can be either Latin or common names but should be consistent throughout the manuscript. Please check the references throughout the whole manuscript.

The manuscript has now been carefully corrected to harmonize the references to organisms.

2. From the provided data (Fig1), Arg133 (R133) is in Walker A not in Site II, could the authors explain it?

The reviewer is right, R133 is indeed located in Walker A. Together with R306 and K316, R133 completes a basic patch on the groove of the RAD51 helical filament. Based on data from *E. coli* RecA and *S. cerevisiae* Rad51, these 3 amino acids are essential for site II function (Cloud et al, 2012). This is also the case for Arabidopsis and is clearly visible in the 3D structure. This is now explicitly mentioned in the introduction (lines 103 to 110). This point has also been raised by reviewer 3 and is now also discussed in the discussion.

3. IU-GUS.8 line is used for somatic homologous recombination assay; blue spots indicate the recombination events. There is no single spot found in 176 *rad51_RAD51-II3A* plants. I was wondering whether the authors checked the *rad51_RAD51-II3A* plants carrying the IU-GUS.8 or not. If so, please provide the data.

As shown in Figure 4C, we have analyzed both *RAD51+/+_RAD51-II3A* and *rad51-/_RAD51-II3A* plants. All data are provided in source data files. These plants were selected based on their homozygosity for both the IU-GUS.8 transgene and the *RAD51-II3A* transgene. We have now reconfirmed the presence of the IU-GUS.8 transgene in these plants by genotyping (through amplification of the IU-GUS.8 transgene) randomly selected seedlings used in the somatic recombination assay. As expected, all seedlings tested were positive for IU-GUS.8 transgene. We also

note that our data are very similar to our previous work with the *rad51* mutant and the RAD51-GFP fusion protein carrying the IU-GUS.8, which showed 1 spot out of 48 plants and 4 spots out of 200 plants, respectively (Da Ines et al., 2013).

Minor points:

4. Line 252: "...5 aligned bivalents at metaphase....", first letter should be capitalized for metaphase.

This has now been corrected (now line 259).

5. Figure 2: The positions and names of three essential amino acids should be written next to them.

We thank the reviewer for pointing out this oversight. Positions and names of the 3 amino acids are now clearly indicated in Figure 2.

6. Figure 4B: Error bar is missing.

We thank the reviewer for pointing this out. Error bar has now been added for line T1-2. However, we note that error bar is not visible for T1-1 since all replicates were equivalent with 100% of plants being sensitive to MMC. This is now clearly written in the figure caption. Number of plants analyzed and number of biological replicates are now also mentioned.

7. Figure 5B: Scale bar is missing; 5C: the photos for Anaphase I in WT and *rad51_RAD51-II3A* are in Telophase I. Please select the correct photos. And remove "10 μ m" from the photos.

A scale bar has been added to Figure 5B. We have now selected the correct Anaphase I pictures and "10 μ m" removed from the photos in Figure 5C.

8. Figure 6: Remove "10 μ m" from the photos.

We have now removed "10 μ m" from the figure as requested.

9. Figure S2: In S2A it is written as "*rad51_RAD51g*" but in S2B it is "*rad51+RAD51g*", please keep the labels consistent; the error bars are missing S2B.

We have now corrected the labels to be consistent throughout the manuscript and figures. Error bars have been added to Figure S2B.

Reviewer #2 (Comments to the Authors (Required)):

Previous studies in yeasts and humans have shown that the DNA binding Site II of RAD51 is essential for somatic recombination but not for meiotic recombination. In addition, RAD51 with the mutation in DNA binding Site II (RAD51-II3A) showed a dominant negative effect in humans. Basically, this study showed that RAD51-II3A has the same effect in Arabidopsis. The data supported the conclusion well.

We thank the reviewer for her/his positive comments on our manuscript.

(1) I have some difficulties with the genotype of transgenic lines. "AtRAD51-II3A", "RAD51-II3A" and "*rad51_AtRAD51-II3A*" were used. Are they the same? If so, please be consistent. What was the genotype of RAD51 in "*RAD51+/-_AtRAD51-II3A*"? How the genotype was determined?

We thank the reviewer for pointing out this lack of clarity. Indeed, AtRAD51-II3A and RAD51-II3A are the same. The labels have now been updated to be consistent throughout the manuscript and figures and reads AtRAD51-II3A to designate the Arabidopsis RAD51-II3A protein. RAD51+/-_AtRAD51-II3A are plants heterozygous for the endogenous RAD51. This is now stated in line 241. These plants carry a T-DNA insertion interrupting the RAD51 gene in only one chromosome. The second RAD51 copy being functional. *rad51_AtRAD51-II3A* are complete knockout of RAD51 gene with T-DNA inserted within RAD51 in both homologous chromosomes. This *rad51* mutant has been previously described

by described in Li et al. 2004 and extensively used since then. Genotype is determined by PCR genotyping as described in Li et al. 2004.

(2) In Fig. 2: The resolution of the lower panel of Fig.2A needs to be improved. In addition, the Close-up views in Fig. 2B and 2C were not consistent with the views in the square.

We thank the reviewer for pointing this out. We have now tried to increase the resolution of the lower panel of Figure 2A. The close-up views do not exactly reflect the views in the square since the close-up views have been slightly rotated and some RAD51 residues behind the 3 essential aa have been hidden to enhance visualization of the three important amino acids. This explanation is now clearly stated in the figure caption.

(3) In Fig. 4B: Please perform statistical analysis.

We have now performed statistical analysis. This is visible in Figure 4 and stated in Figure caption.

(4) In Fig. S1: Compared with the untreated condition, why the mean intensity and sum intensity of RAD51 foci in T1-1 were lower after MMC treatment. Should they be higher?

While this is an interesting observation, we don't agree that the foci should necessarily be more intense after MMC treatment. Before treatment, a given RAD51 focus reflects RAD51 binding to ssDNA flanking a spontaneous DSB (or possibly other lesions), which might differ in length from that flanking MMC-induced DSB. We note that both T1-1 and T1-2 lines show similar tendencies and values. The important point here is that both lines exhibit increased foci intensity compared to wild-type.

Reviewer #3 (Comments to the Authors (Required)):

The manuscript by Valentine et al. through analysis of Arabidopsis RAD51-II3A separation-of-function mutant showed that the RAD51 DNA-binding site II function is required for the recombination activity, but not for nuclear filament formation. Furthermore, the authors demonstrated that the activity of RAD51 DNA binding site II is unnecessary for meiosis. These conclusions are consistent with previous findings from both yeast and humans, which support the idea that the Site II function is conserved across species. The manuscript is well organized and presented in a clear manner. The finding is pretty novel in plants. I have a few concerns that should be considered by the authors to further improve the manuscript.

We thank the reviewer for her/his positive comments on our manuscript.

1.As described in the manuscript, R133 is located in the Walker A domain, while R306A and K316A are located in the DNA binding site II. Because only one of the three mutated amino acids is located in Walker A, as named DNA binding site II is confusing. The conserved R133 in Walker A is thought to affect the binding and hydrolysis of ATP, is it possible that its mutation may prevent ATP hydrolysis, thus stabilizing the RAD51 presynaptic filament from entering subsequent catalysis. Given the fact that AtRAD51-II3A has retained the DNA binding activity and is defective of homologous recombination and repair, which lead to a separation of function phenotype. As a result, this study highlights that DNA binding site II is essential for RAD51 recombinogenic activity. However, the potential function of the other two sites (R306A and K316A) should be at least discussed.

We thank the reviewer for pointing this out. It is true that R133 is located in Walker A (see also our response to reviewer 1) and R306A and K316A in site II. The rationale for mutating the 3 amino acids is that they form a basic patch on the groove of the filament as previously shown (Cloud et al. 2012). We have thus named our mutant as AtRAD51-II3A to be consistent with Yeast and human RAD51-II3A studies. We agree that the respective contribution of each amino acid is an important point that remains to be demonstrated and has not been assessed in our work. We note that Ito et al. (2020 Nature Com) used a *S. pombe* Rad51 mutant mutated for only R306A and K316A (R324 and K334 in *S.*

pombe) and showed that this mutant is severely defective in DNA strand exchange while exhibiting wild-type levels of ATP hydrolysis. Thus, loss of strand exchange activity in site II mutants cannot be explained by a defect in binding and hydrolysis of ATP. This is an important point to mention and it is now stated in the discussion (lines 286-297).

2.As described from Line 150 to 156, however, the hDMC1 sequence used by Xu et al, 2017 is Q313 rather than K313, and the residue in the structure (PDB 5h1c) is also Q, so is Q visible in the structural superposition in Fig2C.

We thank the reviewer for pointing out this oversight. The presence of Q313 is now clearly written (lines 110 and 155). Q313 is indeed visible in Figure 2. It is now clearly indicated and labelled in the superposition in Fig 2C.

3.Line 207 on pg9, "MMC induces DNA interstrand crosslink adducts and in turn, DNA strand breaks that are repaired by RAD51-dependent homologous recombination. " The author should explain MMC function when it first appears.

This sentence has now been moved to accompany the first mention of MMC (lines 186-188).

4.Line 241 on pg10, "RAD51-II3A mutation does not impair meiotic recombination." delete the period, because the other headings do not have a period; Line 333 on pg14, 15µg/ml with less space in the middle; Line 376 on pg15, "Immunolocalization" The first letter "I" does not need to be capitalized ; Lines 607 and 610 on pg23, 30µg/ml, with less space in the middle; "or seedlings 2h (B) or 8h (C)" B should be bold.

This has now been corrected.

5.Fig1B legend, Walker domains are outlined in magenta instead of pink.

This has now been corrected. We have made the colour of Walker domains consistent between Figure 1A and 1B. Both are now Magenta and this is indicated in the figure caption.

6.In Fig 2B-C, three residues should be marked. The color of the residues corresponding to hRAD51/ScRAD51 should also be described. In Fig5B, Bar is not obvious. The data at the bottom of Fig2A is neither mentioned in the text nor the legend.

Residues are now clearly marked in Figure 2B-C and their corresponding color described in Figure caption. The PAE plot at the bottom of figure 2A is now mentioned in the text (lines 162-164) and in the figure caption.

We have also corrected the scale bar in Figure 5B.

7.The "Homologous" in the title of Table1 should be lowercase.

This has now been corrected.

May 2, 2024

RE: Life Science Alliance Manuscript #LSA-2024-02701R

Dr. Olivier Da Ines
Genétique Reproduction and Développement
CNRS UMR6293
28 Place Henri Dunant
Clermont-Ferrand 63001
France

Dear Dr. Da Ines,

Thank you for submitting your revised manuscript entitled "DNA binding site II is required for RAD51 recombinogenic activity in *Arabidopsis thaliana*". We would be happy to publish your paper in Life Science Alliance pending final revisions necessary to meet our formatting guidelines.

- please be sure that the authorship listing and order is correct
- please add callouts for Figure 3A, B and C

A. FINAL FILES:

B. MANUSCRIPT ORGANIZATION AND FORMATTING:

****It is Life Science Alliance policy that if requested, original data images must be made available to the editors. Failure to provide**

original images upon request will result in unavoidable delays in publication. Please ensure that you have access to all original data images prior to final submission.**

The license to publish form must be signed before your manuscript can be sent to production. A link to the electronic license to publish form will be available to the corresponding author only. Please take a moment to check your funder requirements.

Sincerely,

May 7, 2024

RE: Life Science Alliance Manuscript #LSA-2024-02701RR

Dr. Olivier Da Ines
Genetique Reproduction and Developpement
CNRS UMR6293
28 Place Henri Dunant
Clermont-Ferrand 63001
France

Dear Dr. Da Ines,

Thank you for submitting your Research Article entitled "DNA binding site II is required for RAD51 recombinogenic activity in *Arabidopsis thaliana*". It is a pleasure to let you know that your manuscript is now accepted for publication in Life Science Alliance. Congratulations on this interesting work.

DISTRIBUTION OF MATERIALS:

Again, congratulations on a very nice paper. I hope you found the review process to be constructive and are pleased with how the manuscript was handled editorially. We look forward to future exciting submissions from your lab.

Sincerely,
